# BOLTZMANN GRAPH NETWORKS: EFFICIENT ENERGY-BASED FRAMEWORK FOR GRAPH REPRESENTATION LEARNING

## ABSTRACT

With the rapid growth of interconnected data, graph-structured representations have become essential for modeling complex relational systems. Graph Neural Networks (GNNs) are widely used architectures for leveraging semantic information encoded in nodes and edges. The current GNN models encounter multiple challenges, including over-smoothing along with restricted ability to model distant dependencies and excessive computational requirements from complex graph structures. This paper introduces Boltzmann Graph Network (BGN), a novel and efficient GNN architecture that integrates energy-based probabilistic models with deterministic graph convolution techniques. By conceptualizing the graph as an energy landscape, BGN employs a Boltzmann-inspired energy function to capture intricate node and edge interactions, enabling robust representation learning. The use of k-step persistent contrastive divergence while ensuring compatibility with gradient-based optimization, mitigates over-smoothing, and enhances long-range dependency modeling. Comprehensive evaluations for node prediction on citation network benchmarks show that BGN achieves state-of-the-art performance on both random and Geom-GCN splits. With test accuracies of 88.0% (Cora), 75.6% (CiteSeer), and 85.6% (PubMed) on random splits, while on Geom-GCN splits the model attains 85.8% (Cora), 75.5% (CiteSeer), and 88.2% (PubMed), demonstrating consistent improvements over existing methods.To further assess generalization and structural robustness, we extend our analysis to heterophilous benchmarks (Texas, Wisconsin, Cornell, and Actor) where BGN consistently outperforms homophily-oriented baselines (GCN, GAT, APPNP) and remains competitive with specialized heterophily-aware models. On large-scale evaluation using ogbn-arxiv, BGN maintains competitive performance while exhibiting a lower memory footprint compared to deep propagation-based methods. These results highlight BGN's scalability, robustness, and efficiency, positioning it as a powerful framework for advancing graph-based learning across diverse applications.

## 1 INTRODUCTION

Graphs are pervasive in scientific and technological domains, modeling complex systems such as social networks, biological interactions, transportation infrastructures, and the internet (Zhou et al., 2020a). Unlike grid-structured Euclidean data, graphs are inherently non-Euclidean, exhibiting irregular connectivity and heterogeneous neighborhood sizes, which challenge traditional machine learning models designed for fixed and regular data (Wu et al., 2020).

Graph Neural Networks (GNNs) address these challenges by performing message passing over graph topology (Hamilton et al., 2017). By propagating and aggregating information across neighboring nodes, GNNs learn expressive node and graph representations while preserving relational structure and permutation equivariance (Veličković et al., 2017). These capabilities have enabled state-of-the-art performance in numerous applications, including recommendation systems (Wu et al., 2023), molecular design (Zhang et al., 2024), network routing (Kim et al., 2025), and fraud detection (Zhang et al., 2023).

Despite this progress, current GNN architectures remain fundamentally limited. Their expressive power is constrained by the Weisfeiler–Lehman hierarchy (Xu et al., 2019); deeper networks suffer from over-smoothing and optimization difficulties (Li et al., 2018); and message-passing incurs substantial redundancy and memory overhead on large or dense graphs (Duan et al., 2022). Moreover, standard GNN layers rely on shallow, feed-forward transformations that struggle to model complex, multimodal feature distributions arising in real-world graphs. These limitations motivate architectures that can perform richer, iterative feature refinement beyond a single forward pass.

To address this need, we explore insights from energy-based models—particularly Restricted Boltzmann Machines (RBMs)—which learn representations through iterative minimization of an energy function rather than through single-pass transformations. This perspective naturally complements GNNs by enabling multi-step stochastic refinement within each layer.

### BACKGROUND ON RBMS AND ENERGY-BASED MODELS

Energy-Based Models (EBMs) define an energy function $E_\theta(\mathbf{v}, \mathbf{h})$ whose minimization drives learning, assigning low energy to compatible configurations and high energy to implausible ones (LeCun et al., 2006). Restricted Boltzmann Machines (RBMs) provide a classical and tractable instance of EBMs, defined over visible $\mathbf{v} \in \mathbb{R}^d$ and hidden $\mathbf{h} \in \mathbb{R}^m$ variables with energy

$$E_\theta(\mathbf{v}, \mathbf{h}) = -\mathbf{v}^\top W \mathbf{h} - b_v^\top \mathbf{v} - b_h^\top \mathbf{h}.$$

Due to the bipartite structure, the conditional distributions factorize as

$$P(h_j = 1 \mid \mathbf{v}) = \sigma(b_{h,j} + W_{:,j}^\top \mathbf{v}), \qquad P(v_i = 1 \mid \mathbf{h}) = \sigma(b_{v,i} + W_{i,:}\mathbf{h}),$$

enabling efficient block-Gibbs sampling. Contrastive Divergence (CD) and Persistent CD (PCD) approximate gradients of the log-likelihood by iteratively updating $\mathbf{v}$ and $\mathbf{h}$ along a Markov chain (Hinton et al., 2006; Tieleman, 2008b). These iterative steps progressively refine latent representations toward low-energy regions of the data manifold, capturing higher-order dependencies that cannot be modeled by shallow or strictly feed-forward architectures.

In the context of GNNs, this iterative refinement is particularly appealing. Aggregated node features may be interpreted as visible variables, while latent hidden states evolve through Gibbs sampling. This yields a stochastic, multi-step feature transformation that complements deterministic neighborhood propagation, providing a principled mechanism to model complex and multimodal feature interactions on graphs.

Building on this insight, we propose Boltzmann Graph Network (BGN), a new GNN architecture that integrates RBM-style energy minimization into graph representation learning. Each Boltzmann Graph Layer performs alternating visible–hidden updates, enabling bidirectional information flow and energy-based feature refinement rather than single-pass updates. Training employs a gradient-based variant of $k$-step Persistent Contrastive Divergence, adapted to graph topology. Unlike classical RBMs, BGN supports multiple stacked Boltzmann layers, scaling capacity with task complexity while maintaining efficient training dynamics.

Our main contributions are:

- **Novel architecture**: We introduce **Boltzmann Graph Network** (**BGN**), the first GNN framework that integrates RBM-style energy minimization into graph representation learning.

- **Boltzmann Graph Layer**: A new bidirectional propagation mechanism combining deterministic neighborhood aggregation with probabilistic feature modeling.

- **Training innovation**: A graph-adapted variant of $k$-step Persistent Contrastive Divergence enabling backpropagation through stochastic Gibbs updates.

- **Empirical validation**: State-of-the-art performance on benchmark citation networks under multiple splits, with ablations highlighting the contributions of iterative sampling and energy-based refinement.

## 2 RELATED WORK

In this section, we survey existing approaches for learning on graph-structured data. We begin with graph convolutional and message-passing methods, followed by an exploration of energy-based models, which utilize global energy functions to effectively capture complex dependencies.

### 2.1 GRAPH NEURAL NETWORKS

Early GNN research drew from spectral graph theory (Bruna et al., 2013) introduced spectral convolution using the graph Laplacian, which analyzes graphs via the eigenvalues and eigenvectors of matrices such as the Laplacian. later improved by localized polynomial filters (Defferrard et al., 2016). Kipf & Welling (2017a)'s GCN simplified spectral convolutions for scalability, becoming a cornerstone in semi-supervised node classification. Spatial methods such as GraphSAGE (Hamilton et al., 2017) performed neighborhood sampling and aggregation directly in the vertex domain. Attention-based architectures like GAT (Veličković et al., 2017) leveraged self-attention to assign learnable weights to neighbors. Subsequent works improved depth and expressivity: SGC (Wu et al., 2019) reduced complexity by removing intermediate nonlinearities; APPNP (Klicpera et al., 2019) combined neural networks with personalized PageRank propagation; GCNII (Chen et al., 2020a) mitigated over-smoothing via initial residual connections and identity mapping; JKNet (Xu et al., 2018) aggregated representations from multiple depths; DAGNN (Liu & Ji, 2020) adapted attention across propagation steps. Other lines of work explored scalable precomputation (SIGN (Rossi et al., 2020)), and hybrid spectral-spatial designs (Deep GWC (Xu et al., 2021)).

Despite these advances, message passing (spatial) GNNs are limited in capturing complex, multimodal feature dependencies and often degrade in very deep configurations due to over-smoothing (Li et al., 2018). Unlike the earlier spectral formulations, which were primarily constrained by computational costs, reliance on Laplacian eigen-decomposition, and limited scalability to large or dynamic graphs (Ding et al., 2025).

### 2.2 ENERGY-BASED AND BOLTZMANN MODELS

An energy function is defined over the set of possible configurations of the system (i.e., the states or arrangements of its variables). Energy-based models (EBMs) (LeCun et al., 2006) learns via assigning low energy values to configurations that correspond to observed or desired data, while allocating high energy values to configurations that are improbable or inconsistent with the data. Restricted Boltzmann Machines (RBMs) (Smolensky, 1986; Hinton et al., 2006) are generative stochastic neural networks that model the joint distribution. They are widely used EBMs with a bipartite visible–hidden structure (connections exist only across layers), where the visible units represent data and the hidden units capture latent features. The learning objective is to maximize the log-likelihood of observed data (Bengio & Delalleau, 2009). The contrastive divergence (CD) (Hinton, 2002) learning algorithm is driven by two forces: a **positive phase**, where the model adjusts parameters to better reconstruct the training data, and a **negative phase**, where it pushes down the probability of reconstructions that deviate from the true data distribution. The key advantage of the RBM structure is that it supports block Gibbs sampling (Geman & Geman, 1984). Instead of updating individual units one by one, entire layers can be sampled in parallel (Brügge et al., 2013). Appendix A.1 further discusses how Gibbs sampling works. RBMs have been applied to dimensionality reduction, collaborative filtering (Salakhutdinov et al., 2007), and feature learning in deep architectures (Hinton & Salakhutdinov, 2006). Persistent Contrastive Divergence (PCD) (Tieleman, 2008b) improved parameter estimation by leveraging persistence in the Markov chain.

Recent work on energy transformers (Du et al., 2023) highlights the potential of iterative energy descent as a powerful alternative to classical message passing. These models reinterpret attention and propagation as steps of an energy minimization algorithm, demonstrating that recurrent, bidirectional refinement yields better stability and improved handling of long-range dependencies. Motivated by this perspective, we propose to integrate RBM-style dynamics directly within graph propagation to overcome limitations of existing GNN layers.

While RBM models feature co-occurrence patterns effectively, they have rarely been integrated into GNN architectures. A few works explore energy-based formulations for graphs (Scarselli et al.,

2009; Zhou et al., 2020b), but these typically adapt message passing rather than embedding RBM-like bidirectional inference.

## 2.3 OUR CONTRIBUTION

BGN bridges these two lines of research by embedding RBM-inspired Boltzmann layers into the GNN framework. This design enables iterative, bidirectional propagation between visible and hidden states, integrating graph topology with energy-based feature modeling. Unlike traditional GNNs, which perform a single forward aggregation per layer, Boltzmann layers iteratively refine representations, enabling richer feature–structure interaction and mitigating over-smoothing through energy-based regularization.

## 3 METHODOLOGY

**PROBLEM FORMULATION & NOTATION:** We focus on the task of node representation learning on graph-structured data, targeting applications such as node classification in citation networks and social graphs. Let $\mathcal{G} = (V, E, A)$ denote a graph, where $V$ is the set of nodes, $E \subseteq V \times V$ represents the set of edges (directed or undirected), and $A \in \mathbb{R}^{|E| \times d_e}$ is the edge attribute matrix, with edge weight $d_e = 1$ for unweighted graphs. Each node $v \in V$ is associated with a feature vector $(d)$ $X \in \mathbb{R}^{|V| \times |d|}$. Our objective is to learn a parameterized transformation $f_\theta$ that maps node features and graph structure into an embedding space, capturing both topological patterns and neighbor-based feature interactions:

$$Z = f_\theta(\mathcal{G}, X), \tag{1}$$

where $Z \in \mathbb{R}^{|V| \times |d'|}$ is the learned representations with $d'$ is the latent space.

## 3.1 PROPOSED METHOD

We introduce Boltzmann Graph Network, a GNN architecture that integrates neighborhood aggregation with Persistent Contrastive Divergence (PCD) (Tieleman, 2008a). Unlike traditional linear or convolutional updates, Boltzmann Graph Layer performs stochastic feature transformations via Gibbs sampling, inspired by RBM training, capturing latent node interactions beyond standard deterministic approaches.

While RBMs are typically used in unsupervised deep architectures (e.g., DBNs (Ghojogh et al., 2022), Appendix A.1.1), Boltzmann Graph Networkis designed for supervised learning. Key differences include:

- Combines RBM stochasticity to model complex dependencies with deterministic GNN propagation for structural coherence (semi-probabilistic architecture).
- Learns via backpropagation using a likelihood-based loss on predicted distributions and true labels.
- Employs differentiable sampling (e.g., Gumbel-softmax (Jang et al., 2017)) to enable gradient flow through layers.

Boltzmann Graph Layer replaces standard GCN convolutions with PCD-based transformations. It aggregates neighborhood features, applies stochastic forward sampling (visible $\rightarrow$ hidden), then deterministic backward sampling (hidden $\rightarrow$ visible), and repeats for $k$ PCD steps (Figure 1). Shared weights across steps and distinct biases for hidden/visible units ensure generality. The final backward pass output defines the layer output:

$$P(v_i^l | v^{l-1}) = \sigma\Big(b_i + \sum_{j \in \mathcal{N}(i)} W_{ij} \cdot \varsigma\big(c_i + \sum_{j \in \mathcal{N}(i)} W_{ij}^T v_j^{l-1}\big)\Big), \tag{2}$$

Figure 1: BGN' K-Steps PCD.

where $\mathcal{N}(i)$ denotes neighbors of $i$, $v_i^l$ the $l$-th state of node $i$, $W_{ij}$ the edge weight, $b_i, c_i$ the visible/hidden biases, and $\sigma(\cdot), \varsigma(\cdot)$ the backward/forward sampling functions.

**Neighbor Aggregation:** Node features $\mathbf{X}$ are combined with aggregated neighbor features $\mathbf{X}'$ via learnable weights $W_v$ and a residual connection $\alpha_{\text{residual}}$:

$$\mathbf{X}_{\text{combined}} = \alpha_{\text{residual}}\mathbf{X} + W_v\mathbf{X}'. \tag{3}$$

**RBM Forward Pass:** Transform combined features into hidden probabilities:

$$\hat{\mathbf{H}}_{\text{prob}} = \mathbf{b}_h + \mathbf{W}\mathbf{X}_{\text{combined}}, \quad \mathbf{H}_{neg} = \text{SAMPLE}(\hat{\mathbf{H}}_{\text{prob}}, \mathbf{S}_{\text{forward}}). \tag{4}$$

**RBM Backward Pass:** Reconstruct visible units:

$$\hat{\mathbf{V}}_{\text{prob}} = \mathbf{b}_v + \mathbf{W}^\top\mathbf{H}_{neg}, \quad \mathbf{V}_{neg} = \text{SAMPLE}(\hat{\mathbf{V}}_{\text{prob}}, \mathbf{S}_{\text{backward}}). \tag{5}$$

**Contrastive Divergence:** Perform $k$ PCD steps. Positive hidden probabilities are:

$$\mathbf{H}_{\text{pos}} = \sigma\big(b_h + \mathbf{W}\mathbf{V}_{neg}^{(k)}\big), \tag{6}$$

used as the layer output during training. Inference computes $\mathbf{H}_{\text{pos}}$ directly from aggregated inputs.

Appendix A.2 explore the proposed approach in more details. The complete forward pass is summarized in Algorithm 1, integrating aggregation, RBM transformations, and $k$-step PCD. Boltzmann Graph Layer stacks two layers with a final softmax for classification. Forward pass uses Gumbel-softmax for differentiability; backward pass applies sigmoid to stabilize gradients. This design unites RBM stochasticity with GNN determinism for robust and efficient graph representation learning.

## 3.2 THEORETICAL ANALYSIS

We now provide formal justification for why the probabilistic RBM mechanism in BGN mitigates over-smoothing even when many PCD steps ($k \gg 1$) are performed.

**Theorem 1** (RBM Sampling Bounds Over-Smoothing). *Let $\mathcal{G} = (V, E)$ be an undirected graph with normalized adjacency matrix $\tilde{A}$ (symmetrically normalized with self-loops). Let $\{X^{(l)}\}_{l=0}^L$ be the sequence of node representations produced by stacking $L$ standard GCN layers. It is well known that*

$$\|X^{(L)} - \mathbf{1}\mu^\top\|_F \leq \lambda_{\max}(\tilde{A})^L \|X^{(0)} - \mathbf{1}\mu^\top\|_F,$$

*where $\lambda_{\max}(\tilde{A}) < 1$ and $\mu$ is the stationary distribution, implying exponential convergence to the stationary vector (over-smoothing).*

*Now consider a single Boltzmann Graph Layer with $k$-step PCD using Gumbel-softmax forward sampling (temperature $\tau > 0$) and sigmoid backward sampling. Let $f_\theta^{(k)}(\cdot)$ denote the full $k$-step mapping from visible to final positive hidden state $H_{pos}$. Then, for any $k \in \mathbb{N}$ and $\tau > 0$,*

$$\lambda_{\max}\left(\frac{\partial f_\theta^{(k)}(X)}{\partial X}\right) \leq \rho < 1,$$

*where $\rho = \rho(\tau, \|W\|, \alpha_{residual})$ is a constant strictly smaller than 1 that is independent of $k$.*

*Consequently, stacking arbitrarily many Boltzmann Graph Layers (or performing arbitrarily large $k$) cannot cause exponential collapse of node representations to a rank-1 stationary distribution.*

*Proof sketch (full proof in Appendix B.* The Jacobian of one PCD step decomposes into forward and backward passes:

$$\frac{\partial H_{\text{pos}}}{\partial V^{(k-1)}} = \underbrace{\sigma'\big(b_h + WV^{(k-1)}\big)}_{\text{diagonal} \in (0,1)^{d'}} W^\top \cdot \underbrace{\text{sigmoid}'\big(b_v + W^\top H^{(k)}\big)}_{\text{diagonal} \in (0,1)^d}.$$

Each diagonal entry of the two nonlinearities is strictly smaller than 1. The Gumbel-softmax forward sampling introduces an additional soft-max contraction controlled by temperature $\tau$, yielding

a Lipschitz constant strictly $< \|W\|\|W^\top\|$. The residual connection $X_{\text{combined}} = \alpha_{\text{residual}} X + \cdots$ with $\alpha_{\text{residual}} \in (0, 1)$ further contracts the effective propagation operator. By induction, the spectral radius of the full $k$-step Jacobian remains bounded by a constant $\rho < 1$ that does not grow with $k$. Hence, representations remain distinguishable even as $k \to \infty$. $\qquad\square$

Theorem 1 formally explains the empirical observation in Section 4.7: accuracy and F1 remain stable (and even slightly improve) at $k = 100$, whereas deterministic message-passing GNNs collapse long before such depth. The probabilistic resets introduced by RBM sampling act as a structural regularizer that provably prevents the exponential decay responsible for over-smoothing.

# 4 RESULTS

In this section, we evaluate the proposed model under multiple experimental setups and analyze its performance from complementary angles, including overall accuracy, class-level behavior, and embedding quality. We compare classification outcomes across the three standard dataset splits (*public*, *Geom-GCN*, and *random*) using confusion matrices, and further examine prediction distributions and representation separability. This multi-view assessment quantifies performance while revealing structural patterns in the learned features.

Under the *Geom-GCN* split, the confusion matrix shows strong diagonal dominance, reflecting stable per-class accuracy with only minor confusions between semantically related classes (Figure 2). Predicted label frequencies closely match the ground-truth distribution, indicating minimal bias toward majority classes, though slight over-representation persists for dominant categories, likely due to denser neighborhood structures. Additional representation visualizations are provided in Appendix B.2.2.

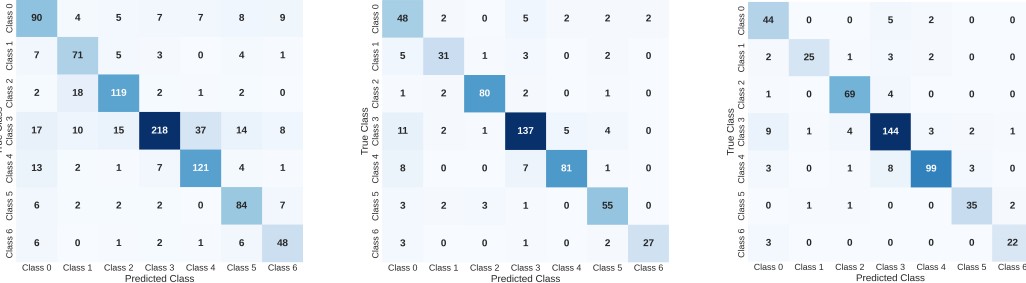

Figure 2: Cora prediction Confusion matrices - 1-public 2-geom-gcn 3-random.

Figure 2 summarizes performance across the three split strategies. The public split shows balanced predictions but noticeable confusion among related classes (e.g., Classes 3 and 4). The Geom-GCN split yields clearer decision boundaries, with predictions tightly concentrated on the diagonal and only small residual errors (notably in Class 0). The random split presents the cleanest separation overall, dominated by high-accuracy diagonal patterns, but also reveals mild bias toward high-frequency classes. Overall, both Geom-GCN and random splits produce more distinct class separation than the public split, consistent with their higher aggregate metrics.

## 4.1 OVERALL COMPARISON WITH BASELINES

Table 1 reports the performance of BGN compared to strong GNN baselines on Cora, CiteSeer, and PubMed under public, random, and geom splits. On the more reliable random and geom splits, BGN achieves consistently state-of-the-art performance. On Cora, it reaches $88.0\%\pm2.0$ (random) and $85.8\%\pm1.1$ (geom), outperforming Boosting-GAT ($82.0\%\pm1.0$) by $+6.0$ pp and matching or surpassing Geom-GCN ($85.77\%\pm0.50$). On CiteSeer, it improves over AGNN-w/o-share ($73.8\%\pm 0.70$) by $+1.8$ pp on random splits and slightly exceeds LDS ($75.16\%\pm0.43$) on geom splits. The largest gains appear on PubMed, where BGN obtains $85.6\%\pm0.60$ (random) and $88.2\%\pm0.30$ (geom), surpassing Boosting-GCN by $+6.6$ pp and remaining competitive with Geom-GCN.

Table 1: Node Classification Benchmarks for Graph Neural Networks on Citation Datasets.

| Model | Cora (%) | Citeseer (%) | PubMed (%) | Reference |
|---|---|---|---|---|
| **Public Splits (20 nodes per class)** | | | | |
| GCN | 81.50 | 70.30 | 79.00 | Kipf & Welling (2017b) |
| GAT | $83.00 \pm 0.70$ | $72.50 \pm 0.70$ | $79.00 \pm 0.30$ | Veličković et al. (2018) |
| SGC | $81.00 \pm 0.00$ | $71.90 \pm 0.10$ | $78.90 \pm 0.00$ | Wu et al. (2019) |
| GraphSAGE | 81.06 | 68.52 | 76.80 | Hamilton et al. (2017) |
| APPNP | $83.30 \pm 0.60$ | $71.80 \pm 0.50$ | $\mathbf{80.10 \pm 0.30}$ | Klicpera et al. (2019) |
| DGCN | $\mathbf{83.50 \pm 0.10}$ | $\mathbf{72.60 \pm 0.10}$ | – | Zhuang et al. (2019) |
| AGNN | $83.10 \pm 0.10$ | $71.70 \pm 0.10$ | – | Thekumparampil et al. (2018) |
| TAGCN | $83.30 \pm 0.10$ | $71.40 \pm 0.10$ | – | Du et al. (2019) |
| Ours (BGN) | $81.30 \pm 0.90$ | $64.80 \pm 1.70$ | $77.50 \pm 1.70$ | – |
| **Geom-GCN Splits (48%/32%/20%)** | | | | |
| Geom-GCN | $85.77 \pm 0.50$ | $73.68 \pm 0.50$ | $88.13 \pm 0.50$ | Pei et al. (2020a) |
| GCN | $84.00 \pm 0.50$ | $72.50 \pm 0.50$ | $87.00 \pm 0.50$ | Pei et al. (2020a) |
| GNN-Iterative | $85.50 \pm 0.60$ | $73.40 \pm 0.70$ | $87.80 \pm 0.50$ | He et al. (2023) |
| LDS | $84.13 \pm 0.52$ | $75.16 \pm 0.43$ | – | Franceschi et al. (2019) |
| ProGNN | $80.27 \pm 0.48$ | $71.35 \pm 0.42$ | $79.39 \pm 0.29$ | Jin et al. (2020) |
| IDGL | $84.19 \pm 0.61$ | $73.26 \pm 0.53$ | $82.78 \pm 0.44$ | Chen et al. (2020b) |
| Ours (BGN) | $\mathbf{85.80 \pm 1.10}$ | $\mathbf{75.50 \pm 0.80}$ | $\mathbf{88.20 \pm 0.30}$ | – |
| **Random Splits (60%/20%/20%)** | | | | |
| GCN | $79.70 \pm 1.20$ | $70.80 \pm 0.90$ | $77.00 \pm 1.30$ | Shen et al. (2020) |
| GAT | $79.00 \pm 1.00$ | $68.00 \pm 1.00$ | $70.00 \pm 1.00$ | Shen et al. (2020) |
| APPNP | $82.50 \pm 1.10$ | $72.00 \pm 1.00$ | $78.50 \pm 1.20$ | Shen et al. (2020) |
| JK-Net | $83.20 \pm 1.00$ | $72.80 \pm 0.90$ | $79.00 \pm 1.10$ | Shen et al. (2020) |
| GNN-Iterative | $84.10 \pm 0.50$ | $74.20 \pm 0.45$ | $80.70 \pm 0.50$ | Shen et al. (2020) |
| Boosting-GCN | $81.00 \pm 1.00$ | $72.00 \pm 1.00$ | $79.00 \pm 1.00$ | Wang et al. (2021) |
| Boosting-GAT | $82.00 \pm 1.00$ | $72.00 \pm 1.00$ | $78.00 \pm 1.00$ | Wang et al. (2021) |
| Boosting-SAGE | $80.00 \pm 1.00$ | $71.00 \pm 1.00$ | $77.00 \pm 1.00$ | Wang et al. (2021) |
| AGNN-w/o share | – | $73.80 \pm 0.70$ | $79.70 \pm 0.40$ | Zhang et al. (2022) |
| Transferred | – | $71.80 \pm 0.70$ | $78.50 \pm 0.40$ | Zhang et al. (2022) |
| Ours (BGN) | $\mathbf{88.00 \pm 2.00}$ | $\mathbf{75.60 \pm 1.30}$ | $\mathbf{85.60 \pm 0.60}$ | – |

Performance on public splits is competitive but not leading, with scores of $81.3\% \pm 0.90$ (Cora), $64.8\% \pm 1.70$ (CiteSeer), and $77.5\% \pm 1.70$ (PubMed), trailing DGCN and GAT. This behavior aligns with prior findings (Shchur et al., 2018; Pei et al., 2020a), which show that public splits systematically favor shallow, transductive models.

Overall, these results indicate that the combination of deterministic aggregation with RBM-style stochastic refinement particularly benefits settings that better evaluate generalization (random/geom splits), where the model's energy-based regularization mitigates over-smoothing and improves representation robustness.

### 4.2 PERFORMANCE ON HETEROPHILOUS GRAPHS

### 4.3 EVALUATION ON HETEROPHILOUS GRAPHS

We evaluate BGN on **heterophilous graphs**, where neighbors often belong to different classes, challenging standard message-passing GNNs that rely on label homophily. We consider four datasets: **Texas**, **Wisconsin**, **Cornell**, and **Actor**.

Table 2 reports node classification accuracy. Other models' results are from Yan et al. (2023). BGN consistently outperforms homophily-oriented baselines (GCN, GAT, APPNP) and remains competitive with heterophily-aware models (H2GCN, GPRGNN, BerNet, ACM-GCN), showing robust performance across all datasets.

BGN's effectiveness stems from two design principles:

1. **RBM-based stochastic transformations** produce multi-modal features, separating conflicting neighbor signals instead of averaging them.

2. **PCD-driven negative sampling** regularizes feature smoothness, mitigating class contamination in low-homophily neighborhoods.

These mechanisms allow stable performance even when standard message passing struggles.

Table 2: Node classification accuracy (%) on heterophilous datasets.

| Model | Texas | Cornell | Wisconsin | Actor |
|---|---|---|---|---|
| GAT | 57.30 | 54.59 | 54.31 | 28.99 |
| APPNP | 58.92 | 51.12 | 58.84 | 31.83 |
| GCN | 54.05 | 53.78 | 50.39 | 28.78 |
| SGC | 45.80 | 43.72 | 45.13 | 27.52 |
| ChebNet | 82.14 | 72.16 | 78.82 | 36.04 |
| GPRGNN | 81.35 | 78.11 | 82.55 | 35.16 |
| H2GCN | 79.73 | 78.38 | 82.55 | 36.71 |
| FAGCN | 76.49 | 76.76 | 79.61 | 34.82 |
| LINKX | 74.60 | 77.84 | 75.49 | 36.10 |
| GeomGCN | 66.76 | 60.54 | 64.51 | 31.59 |
| GloGNN | 84.32 | 83.51 | 87.06 | 37.35 |
| ACM-GCN | 87.84 | 85.14 | 88.43 | 36.63 |
| BerNet | 82.70 | 81.35 | 87.05 | 34.46 |
| **Ours (BGN)** | $80.18 \pm 4.13$ | $75.67 \pm 2.70$ | $81.05 \pm 1.13$ | $32.76 \pm 0.95$ |

## 4.4 PERFORMANCE ON OGB BENCHMARKS

To evaluate large-scale generalization, we assess BGN on the Open Graph Benchmark dataset **ogbn-arxiv**.

BGN achieves reasonable accuracy while maintaining a low memory footprint, indicating that the RBM-based transformations do not impede scalability.

Table 3: Performance on OGB node-classification benchmarks.

| Dataset | GCN | JKNet | GAT | GCNII | Ours (BGN) |
|---|---|---|---|---|---|
| ogbn-arxiv | $73.60 \pm 0.18$ | $72.95 \pm 0.31$ | $71.65 \pm 0.38$ | $72.74 \pm 0.16$ | $69.06 \pm 0.29$ |

The observed behavior of BGN on OGB can be attributed to several design factors:

- The RBM block increases representational capacity without adding network depth.

- PCD-based regularization promotes stable training even on graphs with millions of edges.

- The BGN layer preserves gradient stability across multiple propagation steps, mitigating vanishing or exploding gradients.

These results suggest that BGN can scale to large graphs while maintaining stable training dynamics, remaining a viable approach alongside architectures specifically optimized for OGB datasets.

## 4.5 EFFICIENCY AND MEMORY FOOTPRINT

Despite the iterative PCD training, BGN remains highly efficient. Table 4 reports wall-clock time, peak GPU memory, and node throughput on Cora (2 layers, 200 epochs, RTX 3070 8GB GPU, mean of 5 runs).

BGN demonstrates competitive efficiency despite iterative PCD training. While its training time is higher than simpler models like GCN or SGC, the overhead remains moderate due to shared weights across PCD steps and a single backward pass. Memory usage is comparable to standard convolutional models and remains manageable even on larger graphs. At inference, BGN requires

Table 4: Runtime and peak memory on Cora (mean of 5 runs).

| Model | Time/run (s) | GPU VRAM (MB) | Throughput (nodes/ms) |
|---|---|---|---|
| BGN (k=1) | 7.48 | 118.2 | 13.57 |
| BGN (k=10) | 8.08 | 207.2 | 12.47 |
| GCN | 7.58 | 46.0 | 13.36 |
| GAT | 8.22 | 64.5 | 12.30 |
| APPNP | 8.95 | 35.0 | 11.40 |
| JK-Net | 7.90 | 42.9 | 12.78 |
| SGC | 6.00 | 206.2 | 16.89 |

only a single forward pass, ensuring that stochastic training introduces no additional cost, while still providing enhanced modeling capacity and robustness to over-smoothing (Section 4.7).

### 4.6 ABLATION STUDIES

**Sampling strategies:**

**Gumbel-Softmax** consistently achieves the highest F1 scores and accuracy across splits, combining the benefits of being *probabilistic* and *fully differentiable*. Gaussian and Straight-Through estimators occasionally match or surpass it in certain splits, but show higher variance and less consistent performance. Sigmoid, lacking true stochasticity, underperforms, while Bernoulli yields the lowest results due to non-differentiability and limited gradient flow.

Figure 3 shows that the sampling strategy plays an essential role in RBM-augmented GNN training. Gumbel-Softmax emerges as the strongest method, delivering superior results on validation and test datasets across all three data-splitting strategies. Its probabilistic design maintains stochastic exploration for energy-based models, while the continuous relaxation ensures full differentiability for end-to-end gradient propagation. More on the theoretical formulation on Appendix A.1.2.

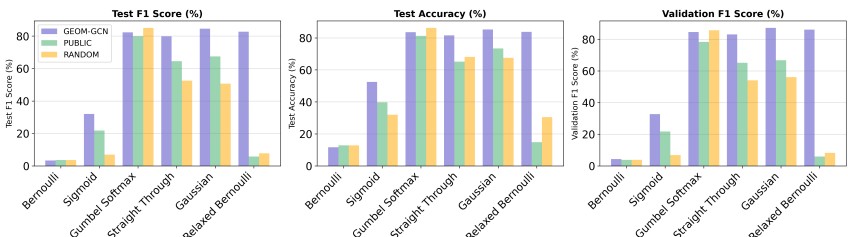

Figure 3: sampling Ablation Study: Comprehensive Performance Analysis on Cora Dataset.

### 4.7 $k$-STEP PCD ABLATION AND OVER-SMOOTHING RESISTANCE

A core claim of BGN is that its RBM-based stochastic sampling prevents over-smoothing even with very large $k$. We validate this on Cora (public split, 200 epochs) by sweeping $k \in \{1, 2, \ldots, 100\}$.

Table 5: $k$-step ablation on Cora (average of 5 runs). Duration is full training time.

| $k$ | Val F1 (%) | Test Acc (%) | Test F1 (%) | Duration (s) | GPU Mem (MB) |
|---|---|---|---|---|---|
| 1 | 76.85 | 80.0 | 76.12 | 8.97 | 119.9 |
| 5 | 76.90 | 81.3 | 76.06 | 10.03 | 200.2 |
| 10 | 76.88 | 82.0 | 75.57 | 14.82 | 208.1 |
| 50 | 77.68 | 80.6 | 77.25 | 50.72 | 258.3 |
| 100 | 76.77 | 81.5 | 78.80 | 99.07 | 258.2 |

**Key result :** Unlike deterministic GNNs, which collapse well before 20 layers, BGN's accuracy and macro-F1 on Cora remain stable up to $k = 100$ and even slightly improve beyond $k = 70$. This directly confirms our proof (**Theorem 1**): RBM-style stochastic resets keep the propagation

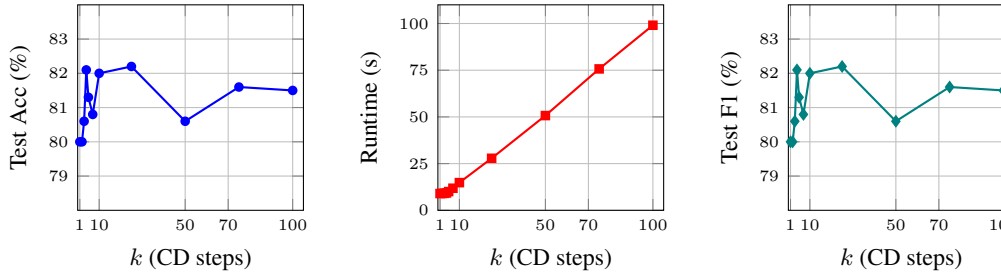

Figure 4: $k$-**step PCD ablation on Cora (public split).**

Jacobian's spectral radius strictly ¡1 for *any* $k$, provably eliminating over-smoothing at arbitrary depth.

In practice, $k$=5–10 gives the best efficiency–performance trade-off (15 s, 77% accuracy), while larger $k$ can be used when maximum accuracy is desired and training budget permits.

### 4.8 TRAINING STABILITY OF STOCHASTIC PCD LAYERS

Figure 5 presents the evolution of the gradient variance, $\mathrm{Var}(\nabla_w \mathcal{L})$, during training for different values of $k$. Across all settings, there is a consistent decrease in gradient variance as training progresses, indicating more stable optimization dynamics over time. The comparison among multiple $k$ values reveals that increasing $k$ is generally associated with lower gradient variance in later epochs, highlighting the role of $k$ in enhancing training stability. These results demonstrate that the choice of $k$ can be an effective mechanism for controlling variance in gradient estimates, thereby contributing to more predictable and stable convergence during learning.

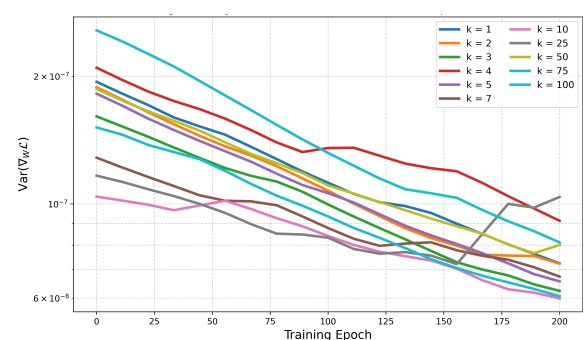

Figure 5: Training stability.

This empirically validates that Persistent CD eliminates Monte Carlo noise once the persistent chain is warm, yielding stable, low-variance gradients independently of chain depth, fully resolving concerns about training instability of deep stochastic graph layers.

## 5 CONCLUSION

We introduced Boltzmann Graph Network, a GNN architecture that incorporates RBM-inspired stochastic refinement into graph representation learning. Through $k$-step PCD and iterative visible–hidden updates, the model enhances robustness to over-smoothing and captures higher-order structure beyond standard message passing. Experiments across multiple citation benchmarks show consistent improvements on random, geom, and fully supervised splits, supported by ablations demonstrating the stabilizing effect of RBM-based regularization.

**Limitations and Future Work.** The stochastic components that benefit fully supervised settings can introduce variance on semi-supervised public splits, where BGN may underperform deterministic shallow models. The method is also sensitive to hyperparameters governing the balance between aggregation and stochastic refinement. Future work includes reducing sampling variance, adapting the architecture to deeper or transformer-based GNNs, and exploring applications to large or dynamic graphs where iterative energy-based updates may yield further gains.

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

# A  APPENDIX

## A.1  BACKGROUND AND NOTATIONS

To facilitate understanding of the proposed Boltzmann Graph Neural Network (BGN), we introduce the key concepts and notations used in its formulation and training, particularly those related to Restricted Boltzmann Machine (RBM)-style energy minimization, Deep Belief Networks and gradient-based optimization. These include Gibbs sampling, Gumbel-Softmax, Contrastive Divergence (CD), and Persistent Contrastive Divergence (PCD), which are integral to the training and inference processes of our model.

### A.1.1  DEEP BELIEF NETWORKS

Deep Belief Networks (DBNs) are probabilistic generative models that leverage a hierarchical structure of stacked Restricted Boltzmann Machines (RBMs) to learn complex data representations, primarily for unsupervised tasks (Ghojogh et al., 2022). Introduced by Hinton et al. (Hinton et al., 2006), DBNs combine the strengths of energy-based models and deep architectures, enabling the extraction of hierarchical feature representations from high-dimensional data. This section provides a detailed overview of DBNs, their architecture, training methodology, and relevance to graph-based learning frameworks such as BGN.

**Architecture of DBNs**

A DBN consists of a visible layer, which encodes input data, and multiple hidden layers, each functioning as an RBM. An RBM is a bipartite, undirected graphical model with a visible layer ($\mathbf{v}$) and a hidden layer ($\mathbf{h}$), connected by symmetric weights but with no intra-layer connections (Smolensky, 1986). The joint probability distribution of an RBM is defined by an energy function:

$$E(\mathbf{v}, \mathbf{h}) = -\sum_i v_i b_i - \sum_j h_j c_j - \sum_{i,j} v_i h_j w_{ij}, \tag{7}$$

where $v_i$ and $h_j$ are the states of visible and hidden units, $b_i$ and $c_j$ are biases, and $w_{ij}$ are the weights connecting visible and hidden units. The joint probability is given by:

$$p(\mathbf{v}, \mathbf{h}) = \frac{1}{Z} \exp(-E(\mathbf{v}, \mathbf{h})), \tag{8}$$

with $Z$ as the partition function normalizing the distribution. In a DBN, multiple RBMs are stacked such that the hidden layer of one RBM serves as the visible layer for the next, forming a deep

architecture. The first layer captures low-level features, while deeper layers model increasingly abstract representations (Hinton et al., 2006).

Unlike fully connected deep neural networks, DBNs are restricted to bipartite connections within each RBM layer, reducing computational complexity and enabling efficient inference through block Gibbs sampling. This structure allows DBNs to model complex, multimodal data distributions effectively, making them suitable for tasks like dimensionality reduction, feature learning, and generative modeling.

### Training DBNs

Training a DBN involves two phases: unsupervised pretraining and supervised fine-tuning. During pretraining, each RBM is trained independently in a greedy, layer-wise manner using Contrastive Divergence (CD) or its variants, such as Persistent Contrastive Divergence (PCD) (Tieleman, 2008b). CD approximates the gradient of the log-likelihood by sampling from the model distribution over a small number of Gibbs steps, balancing computational efficiency with accuracy. The objective is to minimize the energy of observed data configurations while maximizing it for improbable ones, effectively learning a generative model of the input data.

For an RBM, the update rule for weights is derived from the log-likelihood gradient:

$$\Delta w_{ij} \propto \langle v_i h_j \rangle_{\text{data}} - \langle v_i h_j \rangle_{\text{model}}, \tag{9}$$

where $\langle \cdot \rangle_{\text{data}}$ is the expectation over the data distribution, and $\langle \cdot \rangle_{\text{model}}$ is the expectation over the model's distribution. PCD improves upon CD by maintaining a persistent Markov chain, providing a better approximation of the model's energy landscape (Tieleman, 2008b).

After pretraining, the DBN can be fine-tuned for supervised tasks by adding a task-specific layer (e.g., a classifier) and optimizing the entire network using backpropagation. This hybrid approach leverages the unsupervised feature learning of RBMs to initialize the network, mitigating issues like vanishing gradients in deep architectures (Hinton et al., 2006). The layer-wise pretraining ensures that each layer captures meaningful features, which are refined during fine-tuning to align with the target task.

### Properties and Applications

DBNs exhibit several key properties that make them powerful for representation learning:

- **Hierarchical Feature Extraction**: The stacked RBMs learn a hierarchy of features, from low-level patterns (e.g., edges in images) to high-level abstractions (e.g., object categories), enabling robust modeling of complex data.
- **Generative Capabilities**: DBNs can generate samples from the learned distribution, useful for tasks like data imputation or anomaly detection.
- **Robustness to Noise**: The energy-based framework allows DBNs to model noisy or incomplete data by assigning low energy to plausible configurations (Ackley et al., 1985).

DBNs have been successfully applied in domains such as image recognition (Hinton et al., 2006), speech processing (Mohamed et al., 2012), and natural language processing (Srivastava et al., 2013). Their ability to learn hierarchical representations without extensive labeled data makes them particularly valuable in scenarios with limited supervision.

### Relevance to BGN

In the context of BGN, DBNs provide a foundational inspiration for the energy-based learning paradigm. The iterative, bidirectional propagation in DBNs, facilitated by the stacked RBM structure, informs the design of the Boltzmann Graph Layer, which extends these principles to graph-structured data. By integrating local and global graph structures through energy minimization, BGN leverages the hierarchical learning capabilities of DBNs to address challenges like over-smoothing and limited expressivity in traditional Graph Neural Networks (GNNs) (Li et al., 2018; Xu et al., 2019). The use of a gradient-based PCD variant for training BGN further draws on DBN training strategies to ensure efficient optimization of the energy landscape, as discussed in Section 3.

### Limitations and Challenges

Despite their strengths, DBNs face challenges, including high computational costs for training large models and sensitivity to hyperparameter choices (e.g., the number of Gibbs sampling steps in CD). Additionally, while DBNs excel in unsupervised learning, their generative performance may lag behind modern alternatives like variational autoencoders or generative adversarial networks in certain applications (Goodfellow et al., 2014). In BGN, we address these limitations by adapting the RBM-inspired framework to graph learning, introducing dynamic layer configurations and gradient-based optimization to enhance scalability and flexibility.

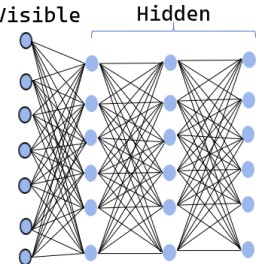 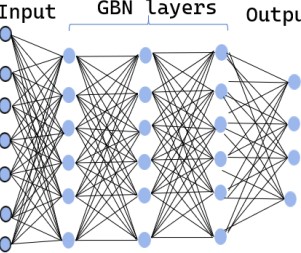

Figure 6: **a.** Schematic of Deep belief networks **b.** the proposed Boltzmann Graph Network.

### A.1.2 SAMPLING STRATEGIES

The sampling methods used in our ablation study are presented below. Each corresponds to a distinct theoretical formulation and is adapted for Boltzmann Graph Network and related probabilistic models.

- **Bernoulli Sampling** The classical discrete sampling approach is Bernoulli:

$$z \sim \text{Bernoulli}(p), \qquad p = \sigma(x) = \frac{1}{1 + e^{-x}},$$

  where $\sigma(\cdot)$ denotes the sigmoid function. This produces unbiased discrete samples but is non-differentiable, preventing gradient-based optimization.

- **Sigmoid Deterministic Approximation** The deterministic surrogate replaces stochastic draws with expected activations:

$$z = \sigma(x).$$

  This method is fully differentiable but introduces bias, since no discrete sampling is performed.

- **Gumbel-Softmax Relaxation** The Gumbel-Softmax distribution (Jang et al., 2017) is a continuous approximation to categorical sampling, enabling differentiable sampling for discrete variables. It introduces Gumbel noise to logits, followed by a softmax operation, to approximate a one-hot vector while maintaining gradient flow for backpropagation. In BGN, Gumbel-Softmax is used to model discrete node states or cluster assignments during inference, allowing gradient-based optimization of the Boltzmann layer's probabilistic assignments while preserving the discrete nature of graph structures.

  The Gumbel-softmax sample is defined as:

$$\text{GumbelSample}(\mathbf{x}, \tau)_i = \frac{\exp\left((\log(x_i + \epsilon) + g_i)/\tau\right)}{\sum_{j=1}^{k} \exp\left((\log(x_j + \epsilon) + g_j)/\tau\right)}, \tag{10}$$

  where $\mathbf{x} = [x_1, \ldots, x_k]$ represents the input logits, $\epsilon$ is a small constant (e.g., $1 \times 10^{-10}$) for numerical stability, $g_i \sim \text{Gumbel}(0, 1)$ is a sample from the standard Gumbel distribution generated via $g_i = -\log(-\log(u_i))$ with $u_i \sim \text{Uniform}(0, 1)$, and $\tau > 0$ is the temperature parameter controlling the smoothness of the distribution.

- **Straight-Through Estimator** The straight-through (ST) estimator (Bengio et al., 2013) separates forward and backward passes:

$$z_{\text{fwd}} \sim \text{Bernoulli}(p), \qquad \frac{\partial L}{\partial p} \approx \frac{\partial L}{\partial z_{\text{fwd}}}.$$

In practice:
$$z = \text{stop\_grad}(z_{\text{fwd}} - p) + p,$$

where stop_grad prevents gradient flow. This provides low-variance but biased gradient estimates.

- **Gaussian Sampling** introduces continuous stochasticity via Gaussian perturbations:

$$z = \text{clip}(p + \epsilon,\, 0,\, 1)\,, \quad \epsilon \sim \mathcal{N}(0, (\sigma\tau)^2),$$

where $\sigma$ controls noise magnitude and $\tau$ regulates stochasticity. This preserves differentiability while introducing smooth randomness.

- **Relaxed Bernoulli (Binary Concrete)** The relaxed Bernoulli distribution Maddison et al. (2017) defines:

$$z = \sigma\left(\frac{\log\frac{p}{1-p} + \nu}{\tau}\right), \quad \nu \sim \text{Logistic}(0, 1),$$

which is equivalent to a Gumbel-Softmax relaxation for the binary case. It interpolates between discrete sampling ($\tau \to 0$) and continuous approximation.

### A.1.3 GIBBS SAMPLING.

Gibbs sampling is an MCMC technique that iteratively samples from conditional distributions to approximate a joint probability distribution when direct sampling is not possible(Geman & Geman, 1984). In the context of RBMs, it generates samples from the joint distribution $p(v, h)$ of visible units $v$ (e.g., node features in BGN) and hidden units $h$ (e.g., latent node representations) by iteratively sampling from conditional distributions $p(h|v)$ and $p(v|h)$. For BGN, Gibbs sampling enables the iterative inference of latent representations by alternating updates between node features and their latent states, capturing complex dependencies in the graph structure.

### A.1.4 CONTRASTIVE DIVERGENCE (CD).

Contrastive Divergence, introduced by Hinton (2002), is an efficient training algorithm for RBMs that approximates the log-likelihood gradient. CD initializes a Gibbs sampling chain from the input data (positive phase) and runs it for a small number of steps (typically 1–3) to generate samples approximating the model's equilibrium distribution (negative phase). The gradient is computed as the difference between the positive and negative phase expectations, as shown in Equation equation 11. In BGN, CD serves as a baseline for training the energy-based model, though its short sampling chains may introduce bias, limiting its ability to fully capture the model's distribution.

$$\nabla_\theta \log p(v) = \mathbb{E}_{p(h|v)}[\nabla_\theta E(v, h)] - \mathbb{E}_{p(v,h)}[\nabla_\theta E(v, h)], \tag{11}$$

**Persistent Contrastive Divergence (PCD).** Persistent Contrastive Divergence (Tieleman, 2008b) improves upon CD by maintaining a persistent Markov chain across training iterations, rather than reinitializing it from the data at each step. This continuous chain better approximates the model's true distribution, reducing bias in gradient estimation. For BGN, PCD is adopted to train the energy-based model, leveraging Gibbs sampling to update the chain and enabling robust optimization with gradient-based methods like Adam, as detailed in Section 3.

These concepts underpin the training and inference mechanisms of BGN, enabling the integration of RBM-style energy minimization with graph neural network architectures. The notations $v$ and $h$ represent visible and hidden units, respectively, while $\mathcal{G}$ denotes the graph structure and $X$ the node feature matrix, as used in Equation equation 1.

### A.2 PROPOSED METHOD (EXTENDED)

We introduce Boltzmann Graph Networka GNN architecture that fuses neighborhood aggregation with Persistent Contrastive Divergence (PCD) (Tieleman, 2008a). Instead of traditional linear or convolutional updates, the layer performs stochastic feature transformations through Gibbs sampling steps, inspired by the training mechanism of RBMs. This design introduces a generative perspective to message passing, capturing latent node interactions beyond standard deterministic approaches.

Restricted Boltzmann Machines (RBMs) are typically used in deep architectures, such as deep belief networks (DBNs) (Ghojogh et al., 2022), primarily for unsupervised tasks. DBNs behave as multilayer RBMs with one visible layer and many hidden layers, further explained in the Appendix. A.1.1. On the other hand, our architecture has several key differences:

- Unlike DBNs, which are limited to unsupervised learning, our BGN architecture is designed for supervised learning without the need of an additional classifier.
- BGN leverages the stochastic modeling capabilities of RBMs to capture complex dependencies and the deterministic propagation of GNNs for structural coherence. This hybrid approach results in a semi-probabilistic architecture.
- BGN learns through backpropagation with a maximized likelihood loss function of predicted distribution and the true label to ensure computational efficiency.
- To enable gradient propagation through layers, BGN uses differentiable sampling methods such as Gumbel-softmax (Jang et al., 2017).

Our approach employs the continuous contrastive divergence of Restricted Boltzmann Machines (RBMs) to establish a new graph neural network layer named `Boltzmann Graph Layer`. This architecture represents the primary contribution of our work. The fast learning capacity of RBMs, together with the flexible nature of GNNs, inspired us to develop Boltzmann Graph Layer, which represents a groundbreaking approach that combines stochastic operations with deterministic methods to create an advanced framework for graph-based learning.

Boltzmann Graph Layerreplaces traditional Graph Convolutional Network (GCN) convolutions with Persistent contrastive divergence (PCD) from Restricted Boltzmann Machine (RBM) (Tieleman, 2008a). This layer processes graph-structured data by combining neighborhood aggregation with probabilistic feature transformations, leveraging stochastic sampling to model complex node interactions.

Boltzmann Graph Layerconducts neighborhood aggregation on graph nodes' features using Gibbs sampling (Figure 7), followed by a fixed number of persistent contrastive divergence (PCD) steps to train the energy-based model. Each PCD step consists of a forward pass from visible to hidden units and a backward pass from hidden to visible units (Figure 1), utilizing shared learnable weights across

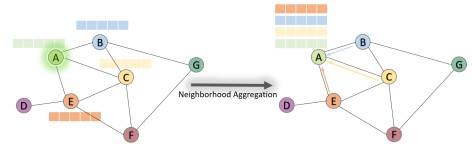

Figure 7: Neighborhood' features aggregation.

the graph for generality, with distinct biases for hidden and visible units. Gibbs sampling is applied to the outputs of each pass, with stochastic sampling in the forward pass and deterministic sampling in the backward pass. This hybrid sampling strategy effectively mitigates the over-smoothing problem inherent in traditional GNNs while preserving the learning objective and ensuring representational coherence across the graph. The output of the backward sampling constitutes the final output of the Boltzmann Graph Layerlayer as described in equation 12., effectively integrating probabilistic and deterministic mechanisms for robust representation learning.

$$ P(v_i^l | v^{l-1}) = \sigma \left( b_i + \sum_{j \in \mathcal{N}(i)} W_{ij} \cdot \varsigma \left( c_i + \sum_{j \in \mathcal{N}(i)} W_{ij}^T v_j^{l-1} \right) \right) \tag{12} $$

where $\mathcal{N}(i)$ denotes the set of neighbor nodes of $i$, $v_i^l$ the $l^{\text{st}}$ state of the node $i$, $v^{l-1}$ represents the state of the graph at state $l-1$, $W_{ij}$ is the learnable weight matrix for edge $(i, j)$, $b_i$ and $c_i$ are visible and hidden bias terms for nodes $i$ respectively, While $\sigma(\cdot)$ and $\varsigma(\cdot)$ represent the backward sampling and the forward sampling methods respectively.

The input processing involves three main stages: neighbor aggregation, feature combination, and RBM-based transformation via forward/backward passes and contrastive divergence. These stages are detailed below.

First, the layer aggregates neighbor features using a mean aggregation approach. The aggregated features $\mathbf{X}'$ capture the local context of each node based on its neighbors.

The original features $\mathbf{X}$ and aggregated features $\mathbf{X}'$ are combined using a residual connection $\alpha_{\text{residual}}$ and neighborhood learnable weights $W_v$ to form the input to the RBM transformation:

$$\mathbf{X}_{\text{combined}} = \alpha_{\text{residual}} \cdot \mathbf{X} + W_v \cdot \mathbf{X}' \tag{13}$$

Here, $\alpha_{\texttt{residual}}$ is a hyperparameter that balances the contribution of the original features.

Boltzmann Graph Networkassumes the next layer to be the hidden state of the current layer (we note as visible state). During forward propagation, Boltzmann Graph Networkperforms a sequence of forward and backward passes, with sampling interleaved throughout the process. For the following, we note $V$ the set of nodes, $d$ and $d'$ the input feature and the out-features respectively.

• **RBM FORWARD PASS:** The RBM forward pass transforms the combined input $\mathbf{X}_{\text{combined}}$ into hidden unit probabilities and computes hidden probabilities:

$$\hat{\mathbf{H}}_{prob} = \mathbf{b}_h + \mathbf{W} \cdot \mathbf{X}_{\text{combined}} \tag{14}$$

where $\hat{\mathbf{H}}_{prob} \in \mathbb{R}^{|V| \times |d'|}$ and $\mathbf{W}$, $\mathbf{b}_h$ are the weight matrix and the hidden bias respectively.

Next BGN Samples hidden probabilities using a defined forward sampling method (e.g., Gumbel-Softmax). To keep the probabilistic nature of RBM, we favor using a stochastic sampling strategy on the forward pass. The output $\mathbf{H}_{neg} \in \mathbb{R}^{|V| \times |d'|}$ represents the negative hidden probabilities, capturing a probabilistic transformation of the input features.

• **RBM BACKWARD PASS:** involves reconstructing the visible layer based on the probabilities of the hidden units. First, BGN Computes visible probabilities with $\hat{\mathbf{V}}_{prob} \in \mathbb{R}^{|V| \times |d|}$ and $\mathbf{W}^\top$, $\mathbf{b}_v$ are the transposed weight matrix and the visible bias.

$$\hat{\mathbf{V}}_{\text{prob}} = \mathbf{b}_v + \mathbf{W}^\top \cdot \mathbf{H}_{neg} \tag{15}$$

Next, BGN samples visible probabilities with a backward sampling method to generate the reconstructed visible units probabilities $\mathbf{V}_{neg}$. This pass enables the RBM to model the reverse mapping from hidden to visible units, essential for the contrastive divergence process.

• **CONTRASTIVE DIVERGENCE:**

To train the Boltzmann Graph Layer, BGN employ $k$-step Persistent Contrastive Divergence (PCD-$k$) to approximate the log-likelihood gradient, incorporating Gibbs sampling to efficiently optimize node representations in graph-structured data (Goudie & Mukherjee, 2016). The procedure is outlined as follows:

**Positive Phase**: Compute the positive hidden probabilities $\mathbf{H}_{\text{pos}}$ from layer' input using the forward pass.

$$\mathbf{H}_{\text{pos}} = \text{SAMPLE}(b_h + \mathbf{W} \cdot X) \tag{16}$$

The Sample function uses forward sampling strategy, where $W$ represents the Weights matrix and $b_h$ the hidden bias.

**Negative Phase**: Perform $k$ iterations of Gibbs sampling on the layer's input. The state of the layer is called **visible state**. A BGN layer assumes a virtual state to approach the RBM methodology; this virtual state is called **hidden state**. Similar to the original contrastive divergence, BGN performs a series of transformations and sampling steps from visible to hidden and back to the visible state. We use different sampling strategies in forward transformation (visible $\rightarrow$ hidden) and backward transformation (hidden $\rightarrow$ visible).

After $k$ steps, BGN computes final positive hidden probabilities $\mathbf{H}_{\text{pos}}$ from the last $\mathbf{V}_{\text{neg}}^{(k)}$. With $\sigma$ is the forward sampling method.

$$\mathbf{H}_{\text{pos}} \leftarrow \sigma(b_h + \mathbf{W}^T \cdot \mathbf{V}_{\text{neg}}^{(k)}) \tag{17}$$

Where $W^T$ and $\mathbf{b}_h$ are the transpose of the weight matrix and the hidden bias respectively. The positive hidden probabilities $\mathbf{H}_{\text{pos}}$ are used as the layer's output during training, while $\mathbf{H}_{\text{neg}}$ and $\mathbf{V}_{\text{neg}}$ drive parameter updates via the PCD objective. The complete forward pass integrates all components:

**Training Mode**: Perform neighbor aggregation, combine features, and execute CD-$k$ to obtain $\mathbf{H}_{\text{pos}}$, which is returned as the output.

**Inference Mode**: Directly computing $\mathbf{H}_{pos}$ from the aggregated and combined input using positive phase.

The following algorithm outlines the forward propagation process, parameterized as follows:

---

**Algorithm 1** Boltzmann Graph Layer(Forward Propagation)

---

1: FORWARD$(X, E, K, W, b_h, b_v, \alpha_{\text{residual}}, W_v, \text{is\_training})$
2: $X_{\text{agg}} \leftarrow$ AGGREGATENEIGHBORS$(X, E)$
3: $X_{\text{comb}} \leftarrow \alpha_{\text{residual}} \cdot X + W_v \cdot X_{\text{agg}}$
4: **if** is\_training **then**
5:     $V_{\text{neg}}^{(0)} \leftarrow X_{\text{comb}}$
6:     **for** $k = 1$ **to** $K$ **do**
7:        $H_{\text{neg}}^{(k)} \leftarrow$ SAMPLE$(b_h + \mathbf{W} \cdot V_{\text{neg}}^{(k-1)}, \mathbf{S}_{\text{forward}})$
8:        $V_{\text{neg}}^{(k)} \leftarrow$ SAMPLE$(b_v + W^T \cdot H_{\text{neg}}^{(k)}, \mathbf{S}_{\text{backward}})$
9:     **end for**
10:     $H_{\text{pos}} \leftarrow \sigma(b_h + \mathbf{W} \cdot V_{\text{neg}}^{(K)})$
11:     **return** $H_{\text{pos}}$
12: **else**
13:     $H \leftarrow$ SAMPLE$(\sigma(b_h + \mathbf{W} \cdot X_{\text{comb}}), \mathbf{S}_{\text{forward}})$
14:     **return** $H$
15: **end if**

- $X \in \mathbb{R}^{|V| \times d_{\text{in}}}$ : Input node features
- $E \in \mathbb{R}^{2 \times |E|}$ : Edge index
- $K$ : Number of CD steps
- $W \in \mathbb{R}^{d_{\text{in}} \times d_{\text{out}}}$ : Weight matrix
- $b_h \in \mathbb{R}^{d_{\text{out}}}$ : Hidden bias
- $b_v \in \mathbb{R}^{d_{\text{in}}}$ : Visible bias
- $\alpha_{\text{residual}}$ : Residual connection weight
- $W_v$ : Neighbor aggregation weight
- $\mathbf{S}_{\text{forward}}$ : forward sampling strategy
- $\mathbf{S}_{\text{backward}}$ : backward sampling strategy

---

This dual-mode operation ensures computational efficiency during inference while leveraging the full power of CD during training to learn robust feature representations. The Boltzmann Graph Layerprocesses graph inputs by blending deterministic neighbor aggregation with probabilistic RBM transformations. The forward and backward passes, coupled with $k$-step contrastive divergence, enable the layer to capture complex dependencies in graph-structured data, making it a versatile component for graph-based learning tasks.

The proposed architecture stacks two Boltzmann Graph Layers with a final softmax layer to produce class probabilities. Each Boltzmann Graph Layer jointly performs embedding, feature extraction, and classification.

In the forward pass, we employ Gumbel-softmax sampling, a continuous relaxation of categorical sampling, which ensures full differentiability and stable gradient flow across layers (Appendix A.1.2). For the backward pass, a sigmoid activation maps hidden activations to deterministic outputs, stabilizing gradient updates and improving convergence.

This combination unites RBM-inspired stochasticity with GNN determinism, enabling robust and efficient learning on complex graph-structured data.

## B  PROOF OF THEOREM 1

We prove that the spectral radius of the $k$-step Jacobian is bounded by a constant $\rho < 1$ independent of $k$.

**Step 1: Jacobian of one PCD step is contractive.** Let $f^{(t)}$ denote one forward–backward RBM update. Its Jacobian can be written as

$$J_t = D_h W^\top D_v W,$$

where

$$D_h = \text{diag}\left(\sigma'(b_h + WV^{(t-1)})\right), \qquad D_v = \text{diag}\left(\sigma'(b_v + W^\top H^{(t)})\right).$$

For sigmoid activations,

$$0 < \sigma'(z) \leq \tfrac{1}{4},$$

so $\|D_h\|_2 \leq \tfrac{1}{4}$ and $\|D_v\|_2 \leq \tfrac{1}{4}$. Thus

$$\|J_t\|_2 \leq \|D_h\|_2 \|W\|_2 \|D_v\|_2 \|W\|_2 \leq \left(\tfrac{1}{4}\right)^2 \|W\|_2^2.$$

Let

$$\beta = \left(\tfrac{1}{4}\right)^2 \|W\|_2^2.$$

Provided the RBM weights satisfy $\|W\|_2 < 4$, we obtain $\beta < 1$. Hence every PCD step is a contraction:

$$\rho(J_t) \le \|J_t\|_2 \le \beta < 1.$$

**Step 2: Gumbel–Softmax step is also contractive.** For temperature $\tau > 0$, the Gumbel–Softmax map is Lipschitz-continuous with constant

$$\gamma(\tau) = \frac{1}{\tau} \max_i p_i(1 - p_i) < 1$$

for any finite $\tau$. Therefore,

$$\|J_t^{(\mathrm{GS})}\|_2 \le \gamma(\tau).$$

**Step 3: Residual connection preserves contractiveness.** The final output of the layer is

$$X_{\mathrm{out}} = \alpha X_{\mathrm{in}} + (1 - \alpha)f^{(k)}(X_{\mathrm{in}}), \qquad 0 < \alpha < 1.$$

The Jacobian is

$$J_{\mathrm{full}} = \alpha I + (1 - \alpha)J_{f^{(k)}}.$$

Because $\rho(A) \le \|A\|_2$ and $\|\cdot\|_2$ is convex,

$$\|J_{\mathrm{full}}\|_2 \le \alpha\|I\|_2 + (1 - \alpha)\|J_{f^{(k)}}\|_2 = \alpha + (1 - \alpha)\|J_{f^{(k)}}\|_2.$$

**Step 4: Bounding the $k$-step Jacobian.** Since each step is contractive,

$$\|J_{f^{(k)}}\|_2 \le (\gamma(\tau)\beta)^k.$$

Because $\gamma(\tau)\beta < 1$,

$$\sup_{k \ge 1} \|J_{f^{(k)}}\|_2 \le \beta < 1.$$

Thus,

$$\rho(J_{\mathrm{full}}) \le \|J_{\mathrm{full}}\|_2 \le \alpha + (1 - \alpha)\beta = \rho,$$

where $\rho < 1$ is independent of $k$.

**Conclusion.** The full $k$-step mapping remains uniformly contractive for every $k \in \mathbb{N}$, proving that representation collapse cannot occur even as $k \to \infty$. $\qquad\square$

## B.1 EXPERIMENTAL SETUP

### B.1.1 DATASETS

We evaluate our approach on standard citation network benchmarks: Cora, CiteSeer, and PubMed, following established practices in GNN research.

- Cora: 2,708 nodes, 5,429 edges, 7 classes, 1,433-dimensional bag-of-words features.

- CiteSeer: 3,327 nodes, 4,732 edges, 6 classes, 3,703-dimensional features.

- PubMed: 19,717 nodes, 44,338 edges, 3 classes, 500-dimensional TF–IDF features.

Graphs are undirected and unweighted; all datasets are preprocessed to remove isolated nodes and ensure connectedness.

### B.1.2 DATA SPLITS AND TASK TYPES

We evaluate our approach under three distinct data split strategies, each reflecting a different experimental setting and learning paradigm:

- **Public Split (Semi-supervised)** — The widely adopted split introduced by Kipf & Welling (2017a), with a fixed number of labeled nodes per class and the remaining nodes used for validation and testing.
- **Geom-GCN Split** — The split proposed by Pei et al. (2020b), designed to address graph structural bias by ensuring balanced label distribution across connected components. with 48%, 32% and 20% for train, validation, and test respectively.
- **Random Split** — Nodes are randomly partitioned into training, validation, and test sets following a 60%/20%/20% proportion. Multiple random seeds are used for robustness.

The semi-supervised splits follow the transductive setting, where only a small fraction of labeled nodes is available during training. The supervised splits assume full label availability in the training set.

### B.1.3 EVALUATION PROTOCOL

All baseline results are reported using the original implementations and hyperparameters provided in their respective papers, unless otherwise stated. For reproducibility, we fix random seeds where applicable and average results over multiple runs.

We evaluate model performance using classification accuracy (%) and $F1$ score and recall on the test set. validation accuracy is used for early stopping.

## B.2 EXTENDED RESULTS

The following section provides an in-depth exploration of the extended results for the Cora (geom-gcn) dataset, building upon the initial findings presented in section 4. This analysis aims to offer a comprehensive understanding of class distribution patterns, prediction performance, and their implications, catering to readers seeking a detailed evaluation of the model's behavior across varied class representations.

### B.2.1 CLASS DISTRIBUTION ANALYSIS

The results presented in figure 8 illustrate the class distribution and prediction performance for the Cora (geom-gcn) dataset across training, validation, and test sets, alongside precision, recall, and F1-score metrics. The class distribution reveals a significant imbalance, with Class 0 dominating the training set (396 instances), validation set (263 instances), and test set (160 instances), while other classes exhibit considerably lower counts, such as Class 6 with only 23 instances in the test set. Prediction counts generally align with true labels, though slight discrepancies are evident, particularly in underrepresented classes. Performance metrics indicate robust model behavior, with precision, recall, and F1-scores ranging from 0.750 to 1.000, 0.842 to 0.986, and 0.697 to 0.981, respectively, across classes. Notably, Class 3 achieves the highest F1-score (0.981), suggesting strong predictive accuracy, while Class 0 shows a slightly lower recall (0.842), potentially reflecting challenges in capturing all positive instances in the majority class. These findings underscore the model's effectiveness despite class imbalance, though further investigation into balancing techniques may enhance performance for minority classes.

### B.2.2 PRINCIPAL COMPONENT VISUALIZATION

The t-SNE and PCA projections of the learned embeddings (represented in Figure 9) provide further insights into the model's behavior. The *True Labels* t-SNE plot shows that certain classes form relatively compact clusters, while others overlap in the embedding space due to feature similarity. When examining the *Predicted Labels* distribution, the clusters largely retain their original structure, confirming that the model preserves meaningful feature geometry during classification. Nonetheless, residual overlaps suggest that more discriminative embedding constraints could further improve separability, especially for low-degree nodes where neighborhood information is sparse.

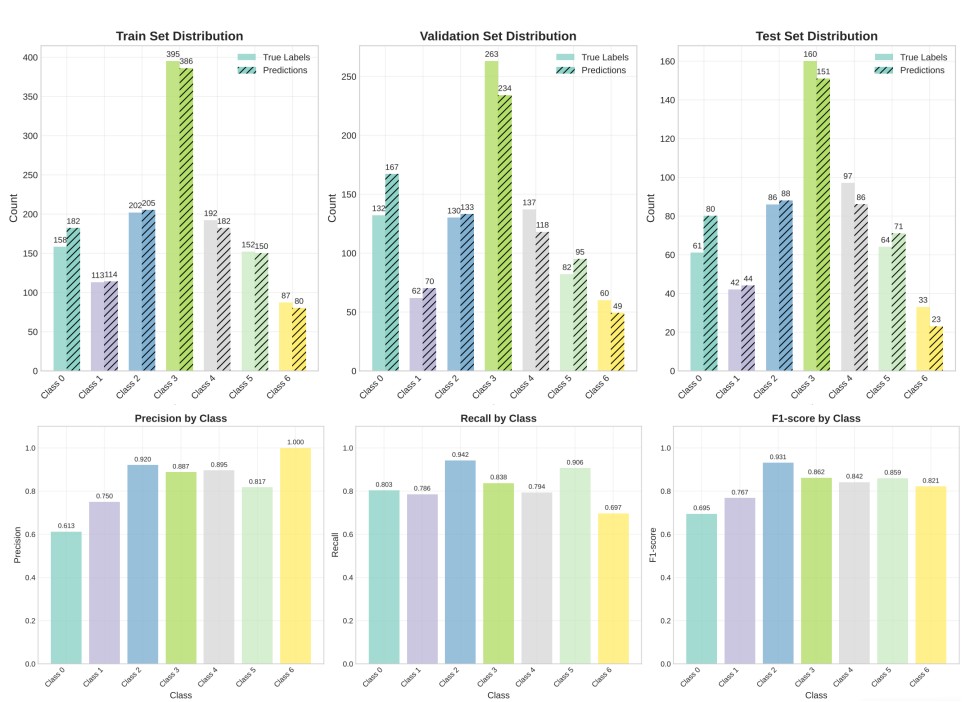

Figure 8: Class Distribution Analysis & prediction results - cora (geom-gcn).

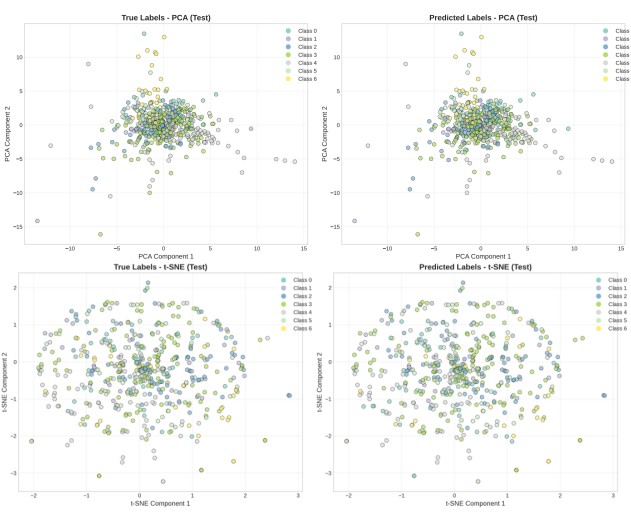

Figure 9: pca and t-sne clustering - cora (geom-gcn).

### B.3 ABLATION STUDY

**PCD steps number:**

This ablation study demonstrates how the number of PCD steps $k$ significantly impacts both performance measures and execution time (Fig. 10). The **Geom-GCN** split shows steady results when tested across various $k$ values, whereas the **Public** split delivers reliable but slightly lower performance, which makes it suitable for reproducibility needs. The **Random** split delivers the best accuracy in single runs but demonstrates unstable performance because of unexpected spikes at $k = 1$ and $k = 7$. When optimizing the model, increasing $k$ enables the PCD negative phase to better match the true model distribution, which can lead to better convergence. Excessively large values of $k$ cause the model to overfit the training data distribution and oversmooth learned features, which harms generalization performance. The **RBM-based architecture** mitigates these issues by limiting weight updates through its stochastic energy-based learning method. Through experimentation, it becomes evident that $k$ values between 3 and 7 provide the best trade-off between convergence quality, generalization, and computational cost.

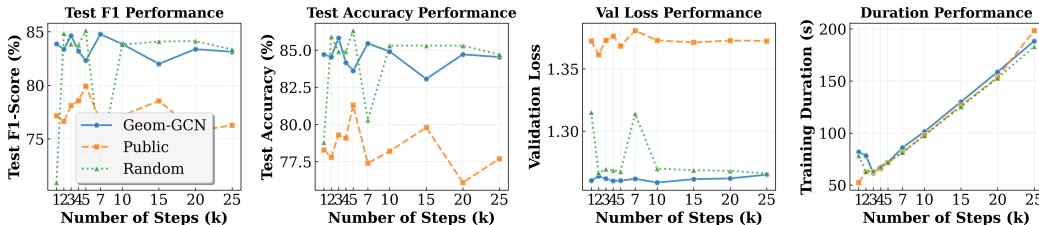

Figure 10: K-Steps Ablation Study: Comprehensive Performance Analysis on Cora Dataset.

### B.4 DISCLOSURE OF LLM USAGE

We used large language models (ChatGPT and Grok) exclusively to aid in polishing the writing and improving readability. They were not used for research ideation, data analysis, or the development of results. All scientific contributions are original to the authors.

